# Simulation Study on Hydrological Process of Soil Cracks in Open-Pit Coal Mine Dump

**Gang Lv** [1,*], **Cong He** [1], **Xinpeng Du** [2] **and Yexin Li** [3]

1   School of Environmental Science and Engineering, Liaoning Technical University, Fuxin 123000, China; hecong1101@126.com
2   School of Geoscience and Surveying and Mapping Engineering, China University of Mining and Technology (Beijing), Beijing 100083, China; duxinpeng1995@126.com
3   School of Architecture and Civil Engineering, Shenyang University of Technology, Shenyang 110870, China; liyexin2008@126.com
*   Correspondence: 18524182288@163.com

**Abstract:** The dumping site is the most serious soil erosion area in an industrial and mining construction area. The development of cracks and water movement in the dumping site is the main factors that induce slope collapse. In this text, the influence of the crack width, rainfall intensity, and two simulation methods of hydrological processes are investigated under artificial rainfall conditions. The results show that the total runoff is affected by two factors, namely rainfall intensity and crack width, and the total runoff decreases with the increase in the crack width. The stable infiltration rate decreases with the increase in the crack width under the same rainfall intensities. When the rainfall intensity is greater than 90 mm/h, the contribution of leakage to the total infiltration is more than 50%. Under simulated rainfall conditions, the total runoff of the solid model was reduced by 5% to 13% compared with the equivalent model. Hence, the cumulative leakage of the solid model is 29% to 71% larger than that of the equivalent model under the same conditions. In this text, the transformation equations from the solid model of the dump site to the equivalent models of runoff, infiltration, and leakage are constructed, and then it can be corrected by the fitting equation.

**Keywords:** crack simulation; water leakage; soil infiltration; slope runoff; 3D printing

## 1. Introduction

Coal mining has driven the local economy, but it has also seriously damaged local production and the living environment [1,2]. The environmental damage caused by large-scale, open-pit coal mining is particularly serious, which causes geological disasters to occur frequently [3]. The dump site is a giant, artificial, unconsolidated accumulation of terraced pagodas with alternating slopes and platforms formed in the process of open-pit mining [4]. Due to the different degrees of compaction, the platform and slope of the dump will have an uneven distribution. A large number of cracks occur on the edge of the platform and are distributed in a band parallel to the edge line of the platform. When rainfall occurs, the slope of the dump pile becomes different from other slopes, and the vadose state is more complicated [5]. Many scholars believe that slope, vegetation coverage, and soil gravel content are the main factors affecting soil erosion on dump slopes [6–12], but they ignore the interception effect of subsidence cracks on runoff. Deep-seated landslides in slopes are often induced by rainfall due to pre-existing cracks or weak layers [13], seriously polluting soil resources and water resources [14]. It greatly affects the runoff and infiltration capacity of the dump slope [15,16]. Due to the existence of cracks, more rainwater will penetrate the interior of the slope through the cracks, and the pore pressure will increase accordingly, affecting the stability of the slope [17–19].

The occurrence of cracks can also greatly affect the water content around the cracks, thereby affecting the migration of water. Consequently, the development of cracks and

the impact on soil water movement have also become the focus of disaster prevention and management in mining areas in recent years [20–23]. In order to explore the effect of cracks on soil hydrology, scholars have carried out many crack simulation methods according to their respective experimental purposes. Hou et al. [24] designed and built an experimental platform for surface channel cracks and flooding alone and deduced the formula for calculating the amount of water intrusion in surface trenches. Bi et al. [25] used the method of collapse to simulate cracks to study the root damage stress caused by mycorrhizae to cracks. Gadi et al. [26] proposed a method to simulate evaporation in soil fissures using a model of root water absorption. In addition, the morphological specificities and development process of cracks have also become the focus of crack research. Wang et al. [27] proposed a method for crack morphology identification and quantification based on data processing and morphological algorithms. Li et al. [28] studied the Soil Water Characteristic Curve and permeability functions of cracked soils at different development stages of dry cracks. Yang et al. [29] studied the crack development law of the working face passing through the valley. Li et al. [30] investigated the crack development process of silty clay in the field. Tjalfe G [31] established a semi-empirical model of soil evaporation under specific crack widths and wind speed conditions.

As far as the current research is concerned, the simulation of cracks and the research on the morphological characteristics of cracks are studied as individual issues [32,33]. The effects of crack morphology on soil hydrology have not been systematically studied by simulating the morphological characteristics of field cracks in laboratory experiments. This study will propose a new crack simulation method that simulates field crack morphology characteristics in laboratory experiments and analyzes the impact of crack morphology on the hydrological process of the dump. This has important theoretical and practical significance for revealing the mechanism of rapid water loss in dumping sites and effectively preventing the occurrence of water and soil loss disasters, such as collapses and landslides caused by soil cracks.

## 2. Materials and Methods

### 2.1. Overview of Test Plots

The test plots are located at the dump site of the Shengli East No. 2 open-pit coal mine east of Xilinhot City, Inner Mongolia Autonomous Region (116°06′41″~116°14′11″ E, 44°02′07″~44°07′05″ N) (Figure 1). The region has a temperate arid and semi-arid monsoon climate, with a mean annual temperature of 1.7~2.5. The mean annual precipitation is 285 mm, of which 70% is concentrated in June–August each year. The mean annual average evaporation is 1795 mm. The soil in the test plots is chestnut soil. The dominant plants are *Leymus chinensis (Trin.) Tzvel*, *Achnatherum splendens (Trin.) Nevski*, and *Stipa baicalensis Roshev*. The total land area of the dump is 7.60 km$^2$, the final disposal elevation is 1156 m, the height of the terrace is 25 m, the width of the platform is 20 m, the slope angle of the step surface is 33°, and the final soil looseness coefficient is 1.15. The soil covering measures the platform and slope mining area (the soil texture is sandy loam), where the thickness of the platform-covering soil is 0.6 m and the thickness of the slope-covering soil is 0.5 m. The artificial vegetation in this area is mainly *Caragana korshinskii Kom*, *Astragalus adsurgens Pall*, and *Medicago sativa L*. The research object selected was the 1105 platform. It was covered with soil and reclaimed in 2013. By 2019, subsidence cracks in a band-like distribution had been generated on the edge (Figure 1).

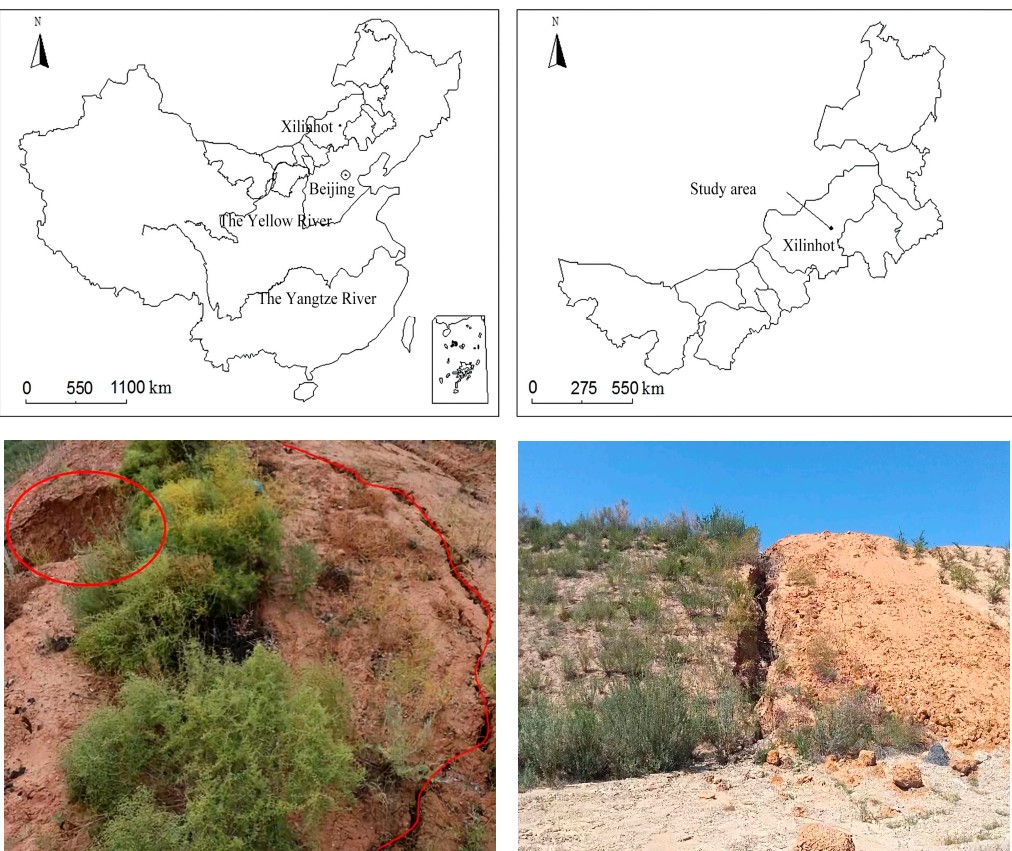

**Figure 1.** Geographical location of the study area and status of cracks and collapses in dumps.

*2.2. Test Methods*

2.2.1. Investigation of Soil Cracks in Dump Site

The 1105 platform was selected as the research object. It was covered with soil and reclaimed in 2013, and a crack investigation was conducted on the platform in August 2019. The main investigation contents were the crack width, length, and quantity, etc. (Table 1). In the platform, covering a total area of 2.5 ha, a total of 48 cracks were found, and the crack lengths ranged from 145 to 6756 cm. The cracks cover a total ground area of approximately 165 m$^2$, of which there are 7 cracks with a width of 3 to 7 cm, 25 cracks with a width of 8 to 12 cm, and 16 cracks with a width greater than 12 cm. See Table 1 for the specific indicators of each crack. Based on the above investigation, cracks with widths of 5, 10, and 15 cm were selected as the research objects.

**Table 1.** Crack index parameters (cm).

| Number | Length | Mean Width | Number | Length | Mean Width | Number | Length | Mean Width |
|---|---|---|---|---|---|---|---|---|
| 1 | 70 | 3 | 17 | 230 | 9 | 33 | 300 | 13 |
| 2 | 580 | 4 | 18 | 215 | 10 | 34 | 1310 | 13 |
| 3 | 430 | 5 | 19 | 140 | 10 | 35 | 1420 | 13 |
| 4 | 520 | 6 | 20 | 280 | 10 | 36 | 210 | 13 |
| 5 | 145 | 7 | 21 | 221 | 10 | 37 | 760 | 14 |
| 6 | 280 | 7 | 22 | 650 | 10 | 38 | 1050 | 14 |
| 7 | 110 | 7 | 23 | 540 | 10 | 39 | 1180 | 15 |
| 8 | 250 | 8 | 24 | 393 | 11 | 40 | 371 | 16 |

**Table 1.** *Cont.*

| Number | Length | Mean Width | Number | Length | Mean Width | Number | Length | Mean Width |
|---|---|---|---|---|---|---|---|---|
| 9 | 190 | 8 | 25 | 512 | 11 | 41 | 2642 | 17 |
| 10 | 570 | 8 | 26 | 280 | 11 | 42 | 400 | 17 |
| 11 | 760 | 8 | 27 | 780 | 11 | 43 | 3195 | 18 |
| 12 | 980 | 8 | 28 | 260 | 12 | 44 | 1680 | 20 |
| 13 | 230 | 9 | 29 | 350 | 12 | 45 | 770 | 24 |
| 14 | 1030 | 9 | 30 | 190 | 12 | 46 | 350 | 32 |
| 15 | 410 | 9 | 31 | 340 | 12 | 47 | 1110 | 34 |
| 16 | 240 | 9 | 32 | 1110 | 12 | 48 | 6756 | 60 |

### 2.2.2. Test Soil

The test soil selected was the platform-covering soil, and the soil type was sandy loam. According to on-site investigations and the determination of indoor physical and chemical properties, the particle composition of sand grains (2~0.02 mm) was 60%, powder grains (0.02~0.002 mm) was 55%, and clay (0.002 mm) was 15%. The soil organic matter content was 3.28%, the total soil nitrogen was 0.12%, the available phosphorus was 1.26 mg/kg, and the available potassium was 87 mg/kg. The soil pH $\approx$ 8, which is weakly alkaline. The bulk density of coal mine slag was 1.89 g/cm$^3$ measured by ring shear testing, and the bulk density of the overburden soil in the dump was 1.35 g/cm$^3$.

### 2.2.3. Indoor Artificial Simulated Rainfall Test

The test system is composed of five parts: SX2004 down-spray artificially simulated rainfall equipment, a self-made glass soil trough, a solid crack model, and sample collection (Figure 2). The water supply equipment consists of a water source, a reservoir, a submersible pump, and a pressure gauge. The rainfall equipment consists of 10 SPRACO circular nozzles, of which the diameter of the nozzles is 2.5 mm and the vertical height of each nozzle is 4.0 m from the ground at a fixed position. The submersible pump is used to boost the pressure of the water in the reservoir. When the test control water pressure is 0.07 MPa, the diameter of raindrops is 2.3 mm, the rainfall uniformity is guaranteed to be above 85%, and the effective rain control area is 20 m$^2$ [34]. During the test operation, the pressure of the nozzle is used to ensure the artificially simulated raindrops obtain a certain initial velocity, and through the control of the rainfall intensity, the kinetic energy of the rainfall is similar to that of natural rainfall. The final speed of raindrops can reach 2.5 mm/s. According to data from local weather stations, the maximum rainfall intensity in the area is approximately 118 mm/h. The rainfall lasted for 60 min in this test, thus the rainfall intensities were set to 60, 90, and 120 mm/h, respectively [35]. Each experimental group was repeated 3 times. According to the three designed rainfall intensities, the rainfall kinetic energy is 23.956, 25.442, and 26.505 J/m$^2 \cdot$s [36].

Figure 2 shows the soil trough is made of plexiglass and is 110 cm long, 50 cm wide, 60 cm high, and 10 mm thick. The vertical section of the internal soil sample filling was a right-angled trapezoid with an upper base of 40 cm and a lower base of 110 cm. The maximum designed filling height is 50 cm. The width of the two crack models is 5, 10, and 15 cm. The slope is 33 according to the design standard of the dump site, and the depth of the crack model is 40 cm, which is the same as the depth of the field cover [37,38]. Before the test, the topsoil samples and slag of the dump site were air-dried. After air-drying, the soil samples were passed through a 2 mm sieve, and the slag was passed through a 5 mm sieve. The bottom layer is 10 cm of slag, and the remaining 10 to 40 cm is air-dried soil, with each 10 cm layer compacted layer by layer. The test controlled the soil bulk density to ensure it reached or approached the design soil bulk density (determined according to field sampling and laboratory tests, wherein the relative error does not exceed 5%). Cracks are divided into two kinds: Solid model and equivalent model. During the process of filling and compaction, the two models were buried in the soil. After the soil

filling was completed, a layer of plastic film was used to cover the surface of the soil tank, and it was allowed to stand for 48 h to be consolidated under the action of gravity.

There were round holes with a radius of 1 cm on the bottom surface of the plexiglass soil trough. There were 16 circular holes evenly distributed in total. The flow out of the circular holes was collected, which is referred to as leakage. The outlet is set at the front edge of the soil trough, which is the runoff volume. After the occurrence of artificial rainfall, the time of runoff and leakage was recorded, and the runoff and leakage samples were collected every 3 min on average and the runoff and leakage were measured. The water balance is infiltration = rainfall-runoff-leakage [39].

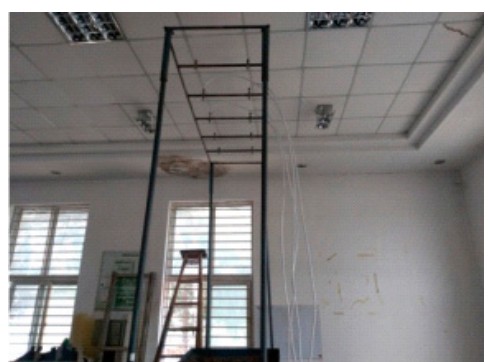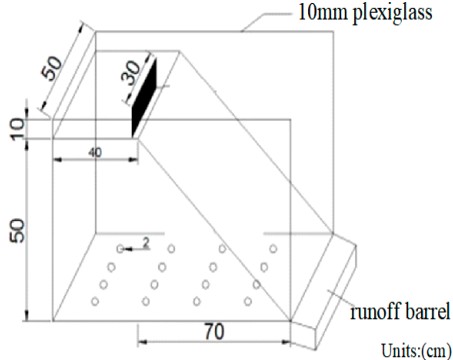

**Figure 2.** Artificially simulated rainfall equipment and schematic diagram of the geometric dimensions of the test soil trough.

### 2.2.4. Equivalent Model Making of Soil Mass Cracks in Dump Site

As shown in Figure 3, the equivalent model is made of a steel plate, which is a triangular prism with the same width and depth as the crack. The equivalent model is simple to make, easy to fill, can directly simulate the width and depth of cracks, and can meet the needs of large-scale crack simulation. However, this method cannot simulate the curvature of the crack and the roughness of the side, and it cannot accurately represent the volume characteristics of the crack.

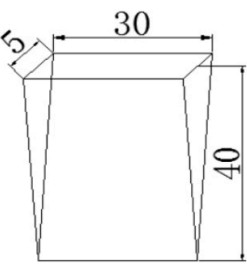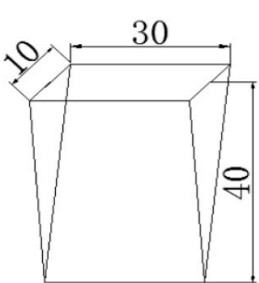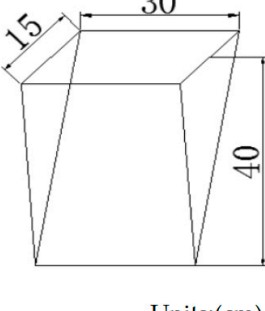

Units:(cm)

**Figure 3.** Schematic diagram of equivalent model.

### 2.2.5. Construction Method of Solid Model of Soil Mass Cracks in Dump Site

In order to express the characteristics of crack curvature and roughness in simulated cracks, a new type of solid model construction for cracks was proposed by combining three-dimensional laser scanning [40], pouring gypsum slurry [41], 3D printing [42,43], and water freezing and thawing [44]. The specific method as shown in Figure 4 is as follows:

(1)　Gypsum Pour Method to Obtain Cracks

Firstly, we poured gypsum into typical three-width soil cracks, took 50 cm in the middle of each crack, and filled it with fine sand on both sides. The ratio of gypsum and water was 1:2 for mixing, and it was stirred evenly for filling and filled until it was flush

with the ground. After 24 h, the crack was solidified and formed, and was separated from the soil. The cracked plaster surface was then wiped clean and the plaster was placed in a foam box for 3D scans. The three-dimensional crack gypsum started to collect three-dimensional information. After the collection, a three-dimensional morphological model of the crack formed, and the index information such as the length, width, depth, volume, surface area, and curvature of the crack was obtained.

(2)    3D Laser Scanning Technology to Collect Information

The test was performed with a Go Scan structured light 3D scanner produced by Creaform. This scanner has high precision and can quickly obtain color, texture, and 3D data of cracks. The collection of three-dimensional information was performed for crack gypsum with three widths. After collection, a three-dimensional morphological model of the crack was formed, and index information such as the length, width, depth, volume, surface area, and curvature of the crack can be obtained.

(3)    3D Printing Technology to Obtain Mold

This test used a Fortus 900 mc industrial-grade 3D printer with a minimum printing thickness of 0.178 mm, and the printing material was polycarbonate (PC). We sprayed a layer of special glue on the specific area that needed to be formed, and evenly sprayed a thin layer of PC powder raw material. When the powder encountered the glue, it quickly fused and solidified. At the same time, the powder remained loose in the area where the glue had not been applied. A layer of glue and a layer of powder were alternately sprayed, and the solid model was "printed" according to the 3D model of the crack. After the process was completed, the excess loose powder was swept away, and the entire crack model was automatically presented. After the crack formed, it was peeled off, cured, and polished.

(4)    The Principle of Water Freezing and Thawing to Simulate Solid Cracks

We sealed the printed mold, then filled the mold with water, put it in cold storage with a temperature of minus 10 °C, and left it for 6 h. After the water was completely frozen, we removed the ice cubes. At this time, the ice cubes were exactly the same as the three-dimensional shape model of the gypsum body. The ice cubes were wrapped and sealed with plastic film to prevent water leakage. At the same time, the upper end was opened and buried in the soil trough. After the ice melted, the water in the plastic film was removed via the siphon method, forming a solid crack that was essentially the same as the crack in the dump.

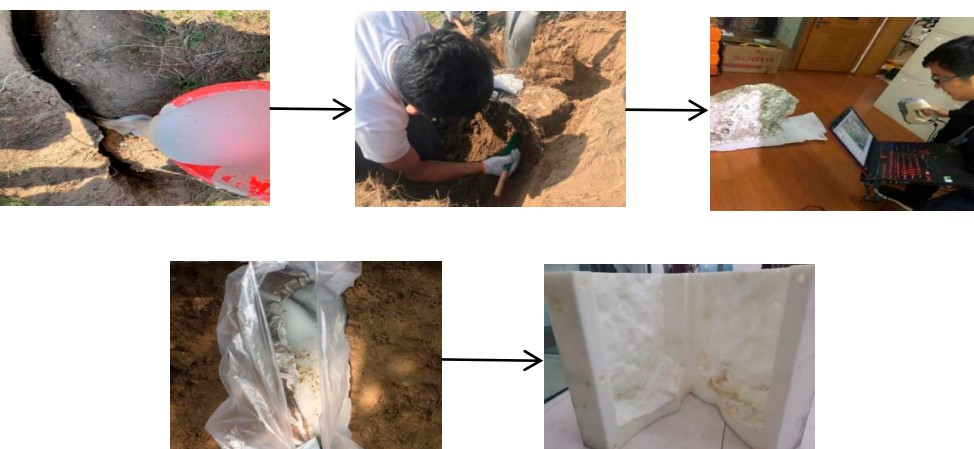

**Figure 4.** Solid model construction flow chart.

By making an equivalent model and a solid model, the parameters of the two models were compared (Table 2). The equivalent model was made of a steel plate with a smooth surface. In the solid model, the rough surface was in contact with the soil, the front and

back were irregular rough surfaces, and the roughness was approximately 1.22. The surface area of the equivalent model was approximately 80% of the solid model, and the volume was only approximately 60% of the solid model. Therefore, compared with the actual situation of field cracks, the difference in morphological characteristics of the equivalent model may greatly affect the runoff and infiltration.

**Table 2.** Comparison of parameters for the two models.

| Crack Number | Model Number | Surface Area/cm$^2$ | Surface Area Ratio | Roughness | Volume/cm$^3$ | Volume Ratio |
|---|---|---|---|---|---|---|
| SC1 | SOM1 | 3054.23 | 1.11 | 1.16 | 5104.48 | 1.70 |
| | EQM1 | 2754.68 | | / | 3000.00 | |
| SC2 | SOM2 | 3497.99 | 1.20 | 1.22 | 11,608.40 | 1.93 |
| | EQM2 | 2918.68 | | / | 6000.00 | |
| SC3 | SOM3 | 4082.97 | 1.32 | 1.27 | 13,742.40 | 1.53 |
| | EQM3 | 3091.82 | | / | 9000.00 | |

Note: SOM is a solid model; EQM is the equivalent model.

## 3. Results and Analysis

### 3.1. Analysis of Slope Runoff Process under the Influence of Solid Model of Soil Cracks in Dump Site

Figure 5 shows the variation law of the runoff volume of the three crack widths under the three rainfall characteristics is similar. The runoff increased at first, and then stabilized. The total runoff was affected by two factors: Rainfall intensity and crack width. Under the same rainfall intensities, the total flow rate of the 15-cm-wide cracks was smaller than 5 cm, which is approximately 85% of the total runoff from 5 cm. The total runoff decreases with the increase in crack width because the larger the crack width, the greater the interception of precipitation. The greater the rainfall intensity of the same crack width, the greater the total runoff. The crack widths were 5, 10, and 15 cm, and the total runoff of the 120 mm/h rainfall intensity was 2.1, 2.2, and 2.3 times that of the 60 mm/h rainfall intensity, respectively. With the increase in crack width, the influence of rainfall intensity on the total runoff also increased.

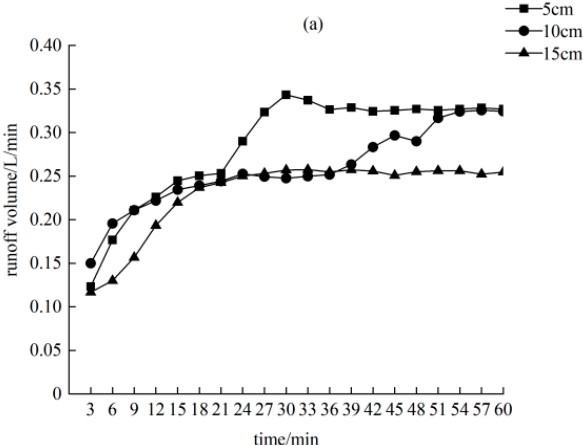

**Figure 5.** *Cont*.

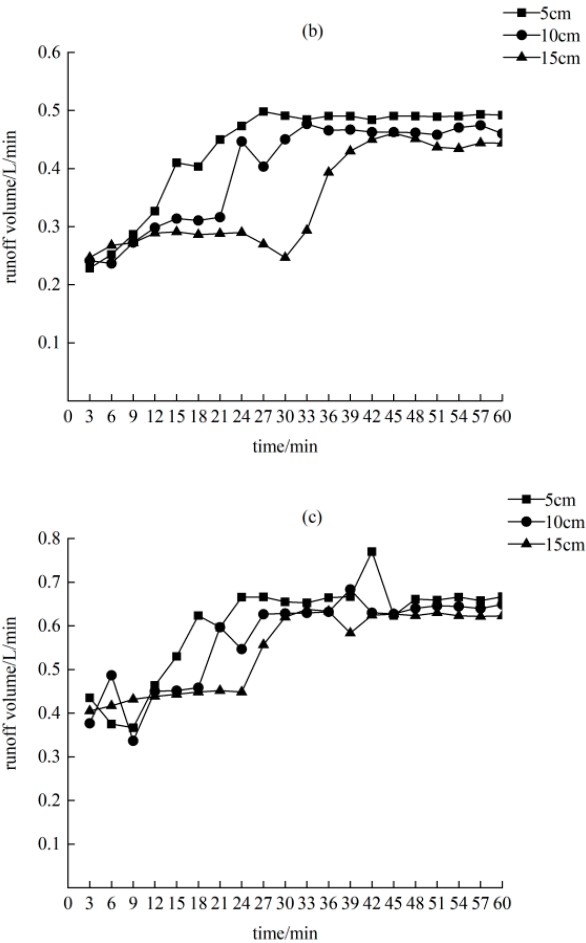

**Figure 5.** Runoff varies with production time. (**a**) Rainfall intensity of 60 mm/h; (**b**) rainfall intensity of 90 mm/h; (**c**) rainfall intensity of 120 mm/h.

The stable flow production is mainly affected by the rainfall intensity, and the effect of the crack width on the stable flow production is only 10~30%. When the crack width is 5 cm, the stable flow production of the 120 mm/h rainfall intensity is twice that of the 60 mm/h rainfall intensity. When the crack width is 10 and 15 cm, the stable flow production of the 120 mm/h rainfall intensity is 2.5 times that of the 60 mm/h rainfall intensity. The above shows that 10 and 15 cm cracks have a greater impact on stable flow production.

*3.2. Analysis of the Infiltration Process under the Influence of Solid Model of Soil Cracks in Dump Site*

The investigation found that the cracks had penetrated the entire overlying soil layer, with the gangue layer at the bottom, and the runoff leaked to the slag layer along the cracks, causing continuous water leakage. Therefore, leakage was studied separately from the infiltration process. By analyzing the soil infiltration water volume per unit of time, defined as the infiltration rate, and the ability of water to pass through the fluid, defined as the leakage rate, two parts of the solid model of soil mass are analyzed.

Figure 6 exhibits that, when a rainfall intensity of 60 mm/h is observed, the initial infiltration rate was 10 > 5 > 15 cm, and the stable infiltration rate was 5 > 10 > 15 cm. The stable infiltration rate of 5 cm was 1.56 times that of the 10 cm rate. The stable infiltration rate increases with rainfall intensity. The initial infiltration rate reached the maximum value in a short period of time. Due to the opening of the soil pores blocked by the soil in the early stage of rainfall, a large amount of infiltration was formed in a short period of time. The leakage rates of the three types of cracks showed an overall trend of increasing first

and then stabilizing. The stable leakage rates were 15 > 5 > 10 cm. In the whole process of infiltration, the cumulative amount of leakage accounted for 30 to 40% of the total amount of infiltration water, and the larger the crack width, the larger the proportion of leakage.

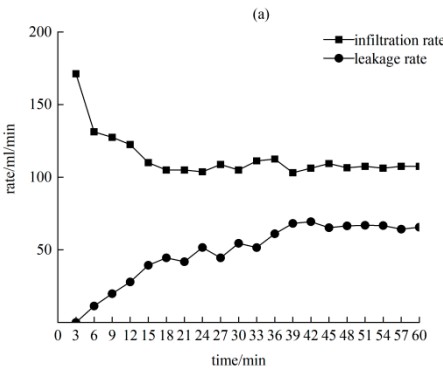

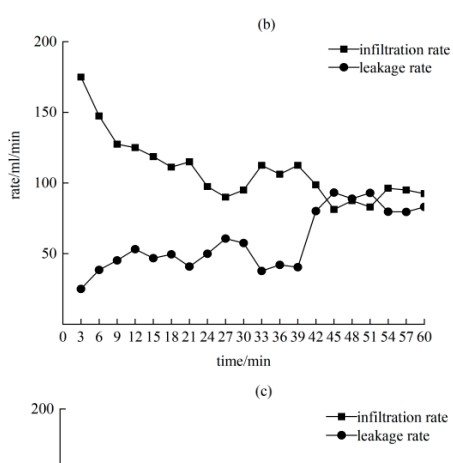

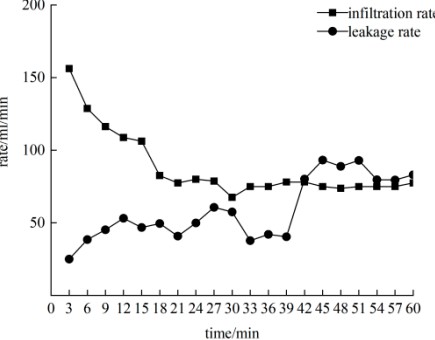

**Figure 6.** Infiltration characteristics under rainfall conditions of 60 mm/h. (**a**) A crack width of 5 cm; (**b**) a crack width of 10 cm; (**c**): a crack width of 15 cm.

Figure 7 exhibits that, with a rainfall intensity of 90 mm/h, the initial infiltration rate was 5 > 10 > 15 cm, and the stable infiltration rate was 5 > 10 > 15 cm. It is shown that the wider the crack, the smaller the stable infiltration rate. The variation law of the leakage rate of the three crack widths showed a trend of increasing first and then stabilizing. The stable leakage rate increased by approximately 20 to 50% with the crack width. In the whole process of infiltration, the cumulative leakage accounted for 47 to 57% of the total amount of infiltration water, and the larger the crack width, the larger the proportion of leakage. When the crack width was 10 and 15 cm, the cumulative leakage accounted for more than 50% of the total infiltration water, which became the main part of the infiltration.

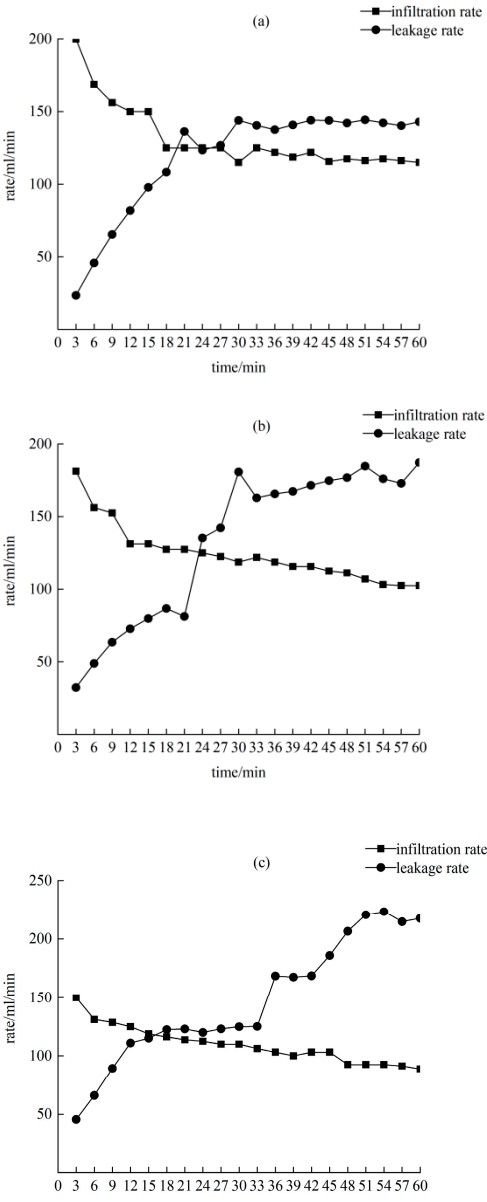

**Figure 7.** Infiltration characteristics under rainfall conditions of 90 mm/h. (**a**) A crack width of 5 cm; (**b**) a crack width of 10 cm; (**c**) a crack width of 15 cm.

Figure 8 exhibits that, with a rainfall intensity of 120 mm/h, the initial infiltration rate was 5 > 10 > 15 cm, and the stable infiltration rate was 5 > 10 > 15 cm. The maximum stable infiltration rate of a 5 cm wide crack was 138 mL/min, which is 1.24 times that of a 15 cm width with the smallest stable infiltration rate. The wider the crack, the lower the stable infiltration rate. The variation laws of the leakage rate of the cracks with the three widths are similar, and the overall trend is to first increase and then stabilize. The stable leakage rate increases by approximately 7 to 25%.

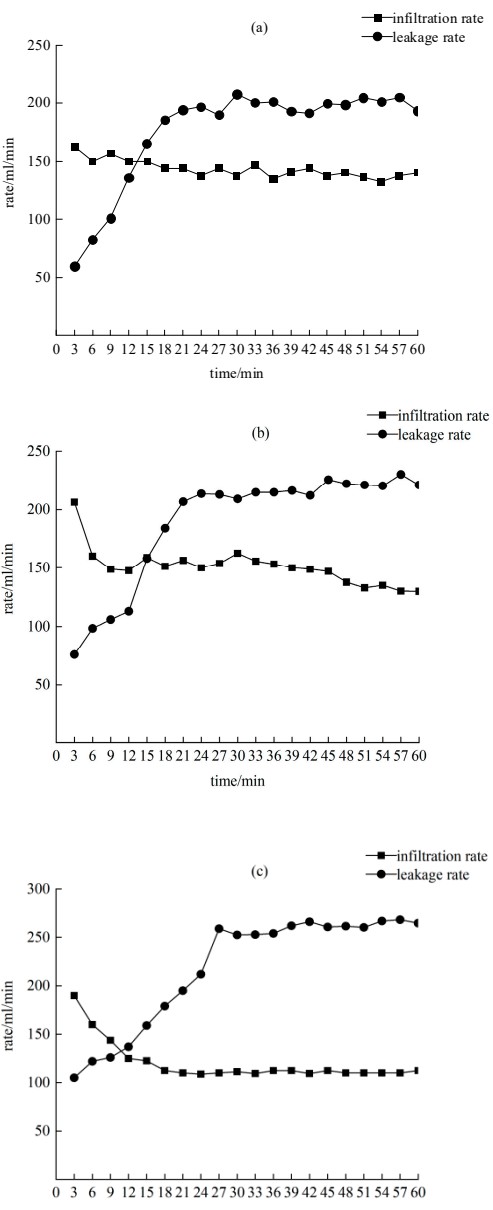

**Figure 8.** Infiltration characteristics under rainfall conditions of 120 mm/h. (**a**) A crack width of 5 cm; (**b**) a crack width of 10 cm; (**c**) a crack width of 15 cm.

In the whole process of infiltration, the cumulative amount of leakage accounted for 55 to 65% of the total infiltration water, and the larger the crack width, the larger the proportion of leakage. At the beginning of the 12 s duration, the leakage rate exceeds the infiltration rate for three crack widths. It showed that under extreme rainfall conditions, the surface layer of the dump platform slope is greatly damaged, which leads to a change in soil porosity. With the increase in rainfall intensity, the stable infiltration rate and stable leakage rate all showed an increasing trend. The stable infiltration rate increased by 24~39% and the stable leakage rate increased by 202~230%. It can be seen that the rainfall intensity has a greater contribution to the leakage rate.

*3.3. Comparative Analysis of the Hydrological Effects of Solid Model and Equivalent Model of Soil Cracks in Dump Site*

The hydrological effects of the solid model and the equivalent model were compared and analyzed, and the internal relationship between the two models was sought in order to

establish a simple crack simulation method to analyze the runoff-infiltration characteristics of soil cracks in the dump site. The results are shown in Figures 9–12.

Figure 9 shows that the variation trend of the total flow production of the two models is similar. The wider the crack, the less total runoff. Under the same conditions, the total runoff of the equivalent model is always higher than that of the solid model. Compared with the equivalent model, the total runoff of the solid model is reduced by 5 to 13%. Under 60 and 90 mm/h rainfall intensities, the solid model was significantly lower than the equivalent model for 5 and 10 cm wide cracks ($p < 0.05$). Under the same crack width, the total flow production of both the equivalent model and the solid model showed that 120 mm/h was significantly higher than 60 and 90 mm/h ($p < 0.05$).

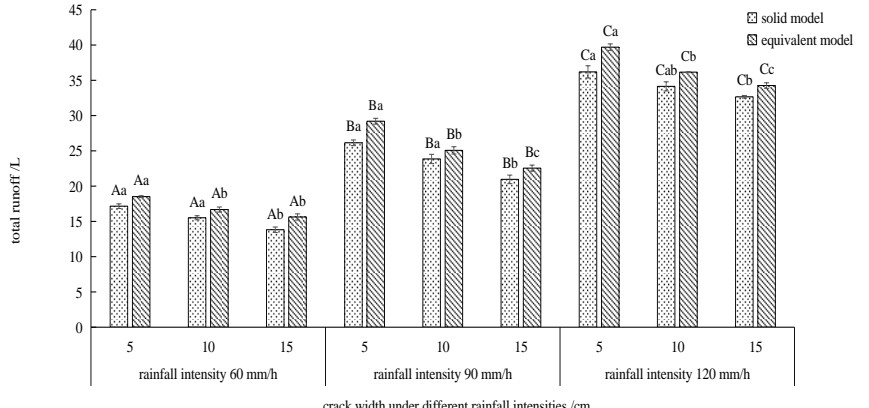

**Figure 9.** Comparison of the total runoff of equivalent model with the solid model. Note: Different lowercase letters indicate significant differences between different crack widths under the same rainfall intensity ($p < 0.05$), and different capital letters indicate significant differences between different rainfall intensities under the same crack width ($p < 0.05$).

Figure 10 shows that the cumulative infiltration amount of the two models decreases with the increase in the crack width, and the same crack width increases with the cumulative infiltration amount with the increase in the rainfall intensity under the same rainfall intensity. Under the rainfall intensity of 60 mm/h, the cumulative infiltration amount of the solid model is always smaller than that of the equivalent model, and the reduction range is between 9 and 16%. The equivalent model of a 15 cm crack is significantly higher than that of the solid fracture model by 16.26% ($p < 0.05$). Under the rainfall intensity of 120 mm/h, the cumulative infiltration amount of the solid model is larger than that of the equivalent model, and the increase is between 3 and 8%. The 15 cm crack solid model is significantly higher than the equivalent crack model by 2.82% ($p < 0.05$), and the cumulative infiltration amount in the 10 and 15 cm crack solid models is 120 mm/h significantly higher than that of 60 and 90 mm/h ($p < 0.05$).

Figure 11 shows that with the increase in the rainfall intensity and crack width, the cumulative leakage of the solid model and equivalent cracks increases. Under the same experimental conditions, the cumulative leakage of the solid model cracks is greater than that of the equivalent model. When the rainfall intensity increases, the cumulative leakage of the solid model increases by 29~71% compared with the equivalent model. Under the rainfall intensity of 120 mm/h, the 15 cm width crack solid model was significantly higher than the equivalent model by 1.60% ($p < 0.05$). Under rainfall intensity of 90 and 120 mm/h, the average increase in the cumulative leakage exceeded 50%. Under 5 and 10 cm crack widths, the cumulative leakage of the equivalent model and solid model showed that 120 mm/h was significantly higher than 60 and 90 mm/h ($p < 0.05$).

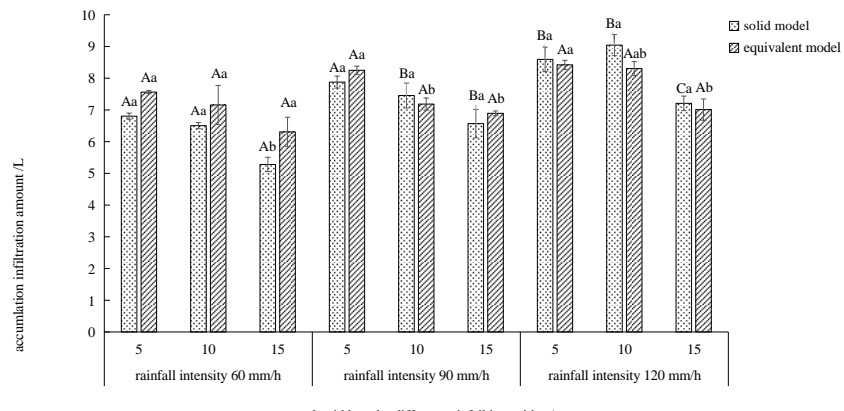

**Figure 10.** Comparison of the accumulative infiltration of equivalent model with solid model. Note: Different lowercase letters indicate significant differences between different crack widths under the same rainfall intensity ($p < 0.05$), and different capital letters indicate significant differences between different rainfall intensities under the same crack width ($p < 0.05$).

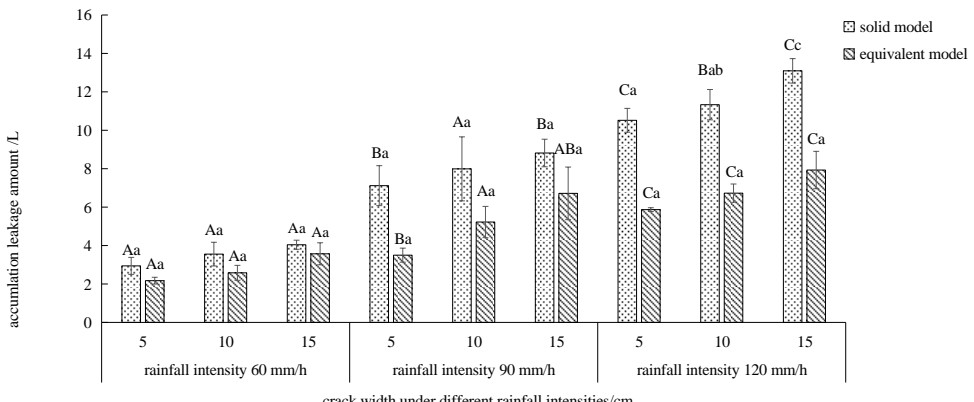

**Figure 11.** Comparison of the leakage between the equivalent model and the solid model. Note: Different lowercase letters indicate significant differences between different crack widths under the same rainfall intensity ($p < 0.05$), and different capital letters indicate significant differences between different rainfall intensities under the same crack width ($p < 0.05$).

Figure 12 shows that under the conditions of different rainfall intensities and crack widths, the total runoff, cumulative infiltration, and cumulative leakage of the equivalent model and solid model are not significantly different, and the variation characteristics are similar. Therefore, an attempt was made to fit the three indicators of total flow production, cumulative infiltration, and cumulative leakage of the two crack simulation methods in order to obtain the transformation equations from the equivalent model to the solid model of runoff, infiltration, and leakage. The results are shown in Figure 12. They all showed a good linear relationship after fitting, and the coefficients of determination were above 0.8. This shows that the total amount of crack production, cumulative infiltration, and cumulative leakage of the equivalent model can be calculated by fitting equations to correct their inaccuracies. The fitting procedures are: $y_1 = 0.94x_1 - 0.35$; $y_2 = 1.42x_2 - 3.34$; $y_3 = 1.68x_3 - 0.55$.

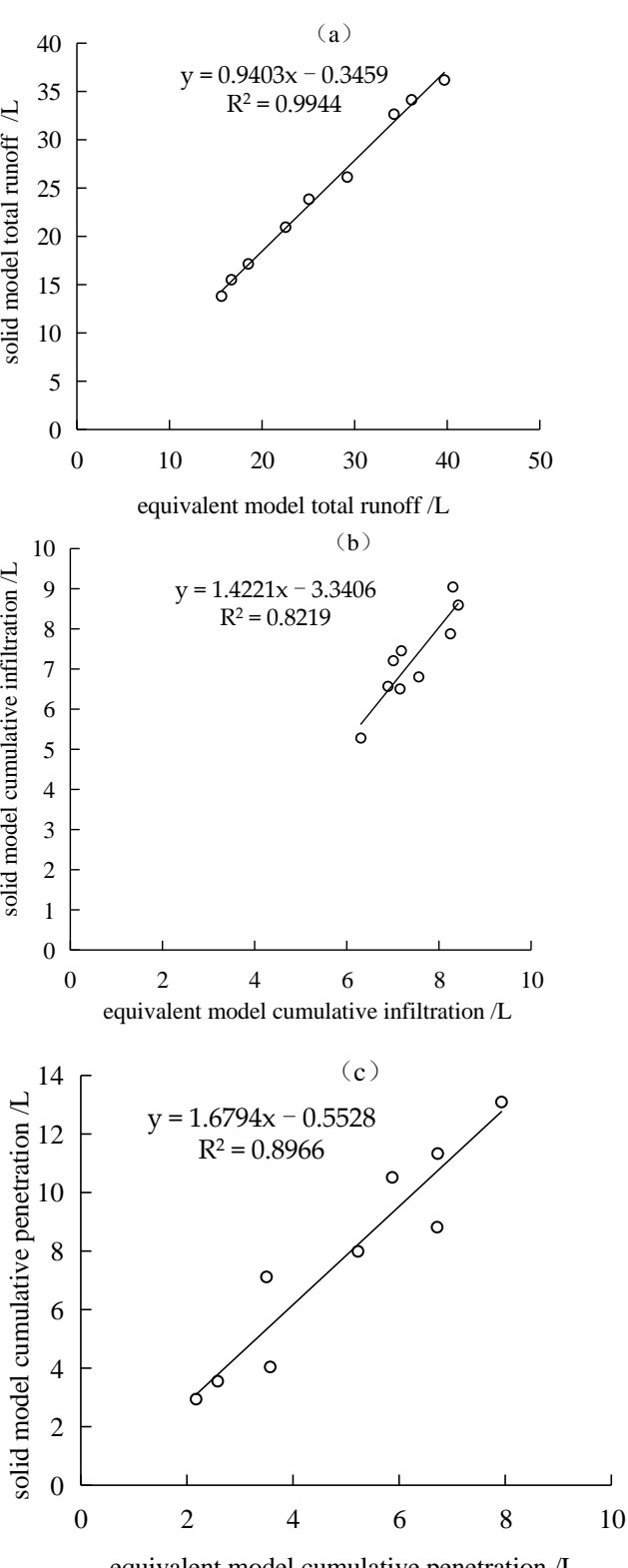

**Figure 12.** The solid model corrects the model of the hydro effect of the equivalent model. (**a**) Runoff fitting model. (**b**) Infiltration fitting model. (**c**) Leakage fitting model.

## 4. Discussion

Due to the different degrees of compaction, the dump platform and slope will experience uneven sedimentation. It is very important to study the influence of cracks on the runoff process of the dump platform and slope under artificial rainfall conditions. The

size of the runoff has become one of the main factors affecting soil erosion [45]. The total amount of runoff from the dump is affected by both the rainfall intensity and the crack width. Under a certain rainfall intensity, with an increase in time, the three kinds of cracks showed a trend of gradually increasing to stable flow production. The total runoff at 120 mm/h was larger than that at 60 and 90 m/h. Due to the increase in water content in the cracks when the dumping site is under extreme rainfall intensity, its permeability coefficient also increases continuously, which causes the water to infiltrate faster in both vertical and horizontal directions and increases its total runoff. With extreme-intensity rainfall, prolonging the duration of light rainfall will significantly increase the soil pore pressure, and the cracked soil slope is more dangerous [46]. Besides, the rainfall intensity mainly affects the runoff process, and the micro-topography will change in a short duration after being hit by a high-intensity rainstorm, causing the runoff to fluctuate significantly in a short period of time. When the rainfall intensity is the same, the total amount of runoff on the slope decreases with the increase in crack width, and part of the rainfall on the surface accumulates in the cracks. The amount of water it accumulates affects the convergence of runoff. It is similar to the impact on runoff under vegetation coverage. Increasing vegetation coverage reduces the slope velocity, slows slope confluence, reduces runoff, and reduces erosion [47]. As the crack width increases in dumps, although the flow production decreases, due to the special mechanism of cracks, the increase in the crack width and depth will cause soil water to move to the depths. The development of cracks will further expand, eventually reducing the stability of the soil and making it more prone to landslides [48]. The stable runoff is more affected by the rainfall intensity than the crack width. In the later stages of the stable runoff, as the water accumulation in the cracks reaches saturation, it will show a stable change trend. At this time, increasing the rainfall intensity will affect the stable runoff.

In the initial infiltration, it is mainly affected by molecular force. During the leakage period, it is mainly affected by capillary force and gravity [49,50]. The stable infiltration period is mainly affected by the combined action of gravity and soil saturation. With the increase in the crack width, the proportion of leakage increases gradually. As the crack width increases, the water storage in the crack increases, and the contact area with the slag layer of the dump also increases, resulting in an increase in leakage. It can be seen that timely control of crack development can effectively reduce the amount of water leakage in dumps under light and moderate rain conditions [51]. Under the rainfall intensity of 90 mm/h, when the crack width is 10 and 15 cm, the cumulative amount of leakage accounts for the total water infiltration and becomes the main part of infiltration. The entire infiltration process does not follow Darcy's law. Since the leakage at the bottom of the crack accounts for more than 50% of the total infiltration, Darcy's law is no longer applicable to soil dumps with cracks.

Due to the difference in roughness and volume between the two models, the solid model has a larger volume and a rougher bottom, and the smooth bottom of the equivalent model has less leakage, thereby increasing the flow production. When the rainfall intensity is 60 mm/h, the cumulative infiltration amount is larger than that of the equivalent model. When the rainfall is 120 mm/h, the cumulative infiltration amount of the solid model is larger than that of the equivalent model. Since the infiltration area of the dump platform model is large, the effect of crack morphology is small, so the effect of the crack morphology difference on infiltration is small. Under rainfall intensities of 90 and 120 mm/h, the average increase in the cumulative leakage exceeded 50%, indicating that the effect of crack morphology and structure on the leakage of soil dumps increases with the increase in rainfall intensity. The method of equivalent cracks simulates soil cracks in loose accumulations [52], which is simple to manufacture and easy to fill, but cannot fully express the surface roughness of the crack and its complex shape. The solid crack model used in this experiment can restore the reducibility of the external form of soil cracks, but the disadvantage is that the operation is more complicated. In order to represent the crack simulation method and strengthen the research on crack hydrological processes more accurately, we

must modify the characteristics of the surface area, volume, roughness, and other indicators of the solid model crack and equivalent model, and the runoff and infiltration results of the equivalent model [53]. Finally, a simple and accurate crack simulation method and runoff-infiltration calculation equation are established.

## 5. Conclusions

Under the same rainfall intensities, the 5, 10, and 15 cm width cracks all showed that the runoff increased first and then stabilized with the passage of time. The total runoff is affected by two factors, namely rainfall intensity and crack width. The total runoff decreases with the increase in crack width under different crack widths. The stable flow production is more affected by the rainfall intensity than the crack width. Under the rainfall intensity of 120 mm/h, the 10 and 15 cm width cracks have the greatest influence on the stable flow production.

Under the rainfall intensities of 90 and 120 mm/h, the stable infiltration rate was 5 > 10 > 15 cm, and the leakage rates of the three crack widths showed an overall trend of increasing first and then stabilizing. Under the rainfall intensity of 90 mm/h, the cumulative leakage of cracks with widths of 10 and 15 cm accounted for more than 50% of the total infiltration water, becoming the main part of infiltration.

Under the same rainfall intensity, the total runoff, cumulative infiltration, and cumulative leakage of the equivalent model and the solid model increased with the increase in crack width and rainfall intensity. Under the same conditions, the total runoff of the equivalent model is always higher than that of the solid model. Compared with the equivalent model, the total runoff of the solid model is reduced by 5% to 13%. The leakage is always larger than the equivalent model, with an increase of 29% to 71%. The simulation of soil dump cracks can correct its inaccuracy through transformation equations from the solid model to the equivalent model of runoff, infiltration, and leakage, and through calculation of the fitting equations.

**Author Contributions:** Conceptualization, G.L. and C.H.; methodology, G.L. and C.H.; software, G.L. and C.H.; validation, X.D.; formal analysis, G.L. and C.H.; investigation, G.L.; resources, Y.L.; data curation, G.L. and C.H.; writing—original draft preparation, G.L. and C.H.; writing—review and editing, G.L. and C.H.; visualization, X.D.; supervision, Y.L.; project administration, G.L.; funding acquisition, G.L. All authors have read and agreed to the published version of the manuscript.

**Funding:** This research was jointly funded by the Liaoning Province "Xing Liao Talents Program" project (Grant. No. XLYC2007046), the National key research and development plan subject (Grant. No. 2017YFC1503105), and the Engineering and Technology Double First-Class Discipline Innovation Team Construction Project of Liaoning Technical University (Grant. No. LNTU20TD-24).

**Institutional Review Board Statement:** Not applicable.

**Informed Consent Statement:** Not applicable.

**Data Availability Statement:** The data presented in this study are available on request from the corresponding author.

**Acknowledgments:** The author wishes to thank the anonymous reviewers for their helpful comments and their careful reading of the manuscript during the revision stage.

**Conflicts of Interest:** The authors declare no conflict of interest.

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
