# Peer review of "Simulation Study on Hydrological Process of Soil Cracks in Open-Pit Coal Mine Dump"

_water, doi:10.3390/w14152302_

Round 1
Reviewer 1 Report
My main concern regarding the current paper is methodology.
First of all, there is no defined slope steepness for the rainfall simulation investigations. The authors mentioned a 30° angle in the introduction, but nothing was given about the set value. This is quite important as we do not know anything about the direction of the artificial cracks related to the slope. Also, no information about the used nozzle, drop spectra, or kinetic energy…
Besides, the used terminology is obscured. We do not know how stable or initial leakage rate was, for example, calculated.
There are functions fitted to three measured points, which is a bit odd.
Figs 5-7 provide data in ml/min despite the highlighted units: mm/min.
There is no mention of replications. However, whiskers on Fig. 8-10 suggest replicated measurements; the results are all about single rainfall simulations. The authors declare that some values are higher than others, even though the whiskers indicate the lack of significant difference. To check this ANOVA or something equivalent would be helpful. On the other hand, if there are no replications, the comparison is quite questionable.
There is no discussion at all. All references cited in the chapter result in a single paragraph that sounds like part of the introduction, which is absolutely independent of the results.
The conclusion chapter simply repeats the results instead of concluding.
My other notes are in the text attached.

Reviewer 3 Report
An interesting manuscript has been prepared; some comments are mentioned below:
The materials and methods in different sections do not have valid scientific references and the authors must provide scientific references for each section. Some articles suggested for use in the materials and methods section:
https://www.sciencedirect.com/science/article/abs/pii/S0169555X19304714?via%3Dihub
https://www.sciencedirect.com/science/article/abs/pii/S0341816219302243?via%3Dihub
https://www.sciencedirect.com/science/article/abs/pii/S0341816219300062
https://www.sciencedirect.com/science/article/pii/B9780128226995000203
They should provide a map of the study area.
Also, a real picture of the laboratory conditions showing the simulator rainfall should be presented in the manuscript.
At the end of the materials and methods section, isn't the statistical analysis section provided? It should be added and statistical analyzes should be used to check the significant difference of the data in different conditions of rainfall intensity and different cracks and presented in the results section of the statistical analysis table and then discussed.
line 172 to 230, it should be transferred to the materials and methods section
Figures 6-7 and 8 should be merged to allow comparison in different rainfall intensities
In Figures 9-10 and 11, statistical analysis should be used to show in each figure whether there is a statistical difference between different states or not?!
In line 171, the title should be only the results and the discussion should be deleted, because the discussion is presented again in line 369, or the results and discussion section should be merged.
Round 2
Reviewer 1 Report
The text is improved, however, it is still not ready for publication.
The authors commented on my notes posted in the text but did not reflect my main concerns.
I am not able to check the changes as they are not tracked in the pdf, and I do not have time to read it over again.
The results are based on single measurements without replications (in the former version, there were whiskers on chart bars suggesting error values of a single measurement!!!). I am afraid that the lack of replications undermines this study's robustness.
The abstract reads: “the total runoff and stable infiltration rate of cracks with a width of 5 cm were the highest”. I cannot imagine how both infiltration and runoff can be the highest in a case as they are inversely related.
The nozzle type, the used pressure value, and kinetic energy are still missing.
Titles: “3 Results and discussion”; “4 Discussion”
Reviewer 3 Report
Congratulations to the authors of the manuscript.
They have considered my comments and revised the manuscript completely, and the manuscript can be published in the Water journal.